# Delegated Proof of Stake Consensus Mechanism Based on Community Discovery and Credit Incentive

**DOI:** 10.3390/e25091320

**Published:** 2023-09-10

**Authors:** Wangchun Li, Xiaohong Deng, Juan Liu, Zhiwei Yu, Xiaoping Lou

**Affiliations:** 1School of Information Engineering, Jiangxi University of Science and Technology, Ganzhou 341000, China; 6720210626@mail.jxust.edu.cn (W.L.);; 2School of Electronics and Information Engineering, Gannan University of Science and Technology, Ganzhou 341000, China; 3Key Laboratory of Cloud Computing and Big Data, Ganzhou 341000, China; 4College of Information Science and Engineering, Hunan Normal University, Changsha 410081, China; louxiaoping@hunnu.edu.cn

**Keywords:** community discovery, DPoS, consensus mechanisms, GN algorithm, PageRank algorithm, blockchain

## Abstract

Consensus algorithms are the core technology of a blockchain and directly affect the implementation and application of blockchain systems. Delegated proof of stake (DPoS) significantly reduces the time required for transaction verification by selecting representative nodes to generate blocks, and it has become a mainstream consensus algorithm. However, existing DPoS algorithms have issues such as “one ballot, one vote”, a low degree of decentralization, and nodes performing malicious actions. To address these problems, an improved DPoS algorithm based on community discovery is designed, called CD-DPoS. First, we introduce the PageRank algorithm to improve the voting mechanism, achieving “one ballot, multiple votes”, and we obtain the reputation value of each node. Second, we propose a node voting enthusiasm measurement method based on the GN algorithm. Finally, we design a comprehensive election mechanism combining node reputation values and voting enthusiasm to select secure and reliable accounting nodes. A node credit incentive mechanism is also designed to effectively motivate normal nodes and drive out malicious nodes. The experimental simulation results show that our proposed algorithm has better decentralization, malicious node eviction capabilities and higher throughput than similar methods.

## 1. Introduction

A blockchain contains technologies such as distributed data storage [1], peer-to-peer data transmission [2], consensus mechanisms [3] and cryptographic mechanisms [4], and it is considered one of the most disruptive innovations in computing after mainframes, personal computers and the internet. It is just because of these core technologies that each node can communicate with each other and finally reach a consensus. In addition, the maintenance and management of the blockchain relies on the participation of all nodes, so it is decentralized. Currently, this has caused new technological innovations and industrial change worldwide. Blockchain technology originated from a person working under the pseudonym of “Satoshi Nakamoto”, who proposed a famous digital currency called Bitcoin in 2008. In particular, it enables untrusted entities to conduct transactions without an intermediary, reducing transaction costs and making transactions transparent. With the success of Bitcoin, blockchain was first applied in the field of digital finance and was represented by digital currencies. Afterward, the emergence of Ethereum [5] expanded the blockchain from a single finance field to broader fields such as e-government [6], education [7], healthcare [8], the Internet of Things [9], and energy [10].

The characteristics of a blockchain have attracted the attention of some investors. For example, Böyükaslan et al. [11] conducted a multifaceted study on investors’ decisions on cryptocurrencies and found that security is the most important factor considered by investors. In addition, the sub-factors are digital signatures and strong electronic encryption technology. Therefore, the higher the security of the cryptocurrency, the more investors it will attract. In short, the cryptographic mechanism of a blockchain has been recognized by people. Oshodin et al. [12] reviewed and organized the literature on digital currencies and found that some relevant characteristics of people, such as user emotion, social communication, and trust issues, can have a great impact on the value and adoption of digital currencies. Zarifis et al. [13] conducted a survey of consumers and found that the low transaction costs of electronic currencies can be very useful to retailers, while government participation and regulation can have a positive effect on people’s transactions in digital currencies. For example, the Bitcoin system that applies blockchain technology has achieved great success.

In addition to the above mentioned, blockchain governance has also attracted attention from some researchers. For instance, Pelt et al. [14] designed a conceptual framework to improve researchers’ understanding of blockchain governance, which provided a structured way for companies, stakeholders, and regulators to analyze how to carry out blockchain governance so that blockchain projects can be sustained. Cowen [15] adopted the concept of a “legislator” in new institutional economics and political theory to apply blockchain governance to private governance. In particular, blockchain technology can reduce transaction costs in the absence of trust so that political entrepreneurs can use the idea of governance to credibly alienate some power. Alston [16] found that the blockchain ecosystem resembles a kind of constitutional rule set, and the modification of the data structure on the blockchain resembles the design and revision of a constitution. Thus, understanding the design and revision of constitutions can facilitate the development of blockchain governance. At the same time, the unique governance solutions generated by the competition of blockchain applications on cryptocurrencies also provide informative insights for automated governance in the public sector. Lumineau et al. [17] found that blockchain adopts a cooperative and coordinated approach to governance, which is different from traditional contract and relationship governance. Uniquely, they analyzed the organized collaborations of blockchain governance from the perspective of organizational science, providing organizational scholars with a new perspective on the study of organizing collaborations. Romme et al. [18] divided the life cycle of the blockchain system into three stages. In addition, they also studied the characteristics of the blockchain governance mechanism at each stage and gave corresponding recommendations, which improves the sustainability of the blockchain ecosystem for the future.

The consensus algorithm [3], as one of the core technologies of a blockchain, has a great impact on performance indicators such as latency, transaction throughput, and the block generation rate of blockchain systems. In recent years, there have been many types of consensus mechanisms. Generally, there are three types of consensus algorithms: the first is proof of work (PoW) [19], based on computing power competition; the second is proof of stake (PoS) [20], based on CoinDays; and the third is proof of credit (PoC) [21], based on credit. In particular, delegated proof of stake (DPoS) [22], an improved consensus mechanism derived from PoS, has become a mainstream algorithm. Specifically, it elects *k* representative nodes by voting, and when each round of elections is over, these *k* nodes take turns generating blocks. Compared to PoW and PoS, the DPoS algorithm reduces the waiting time required for transaction verification and increases the block generation rate by electing representative nodes to generate blocks [23,24]. Some researchers have proposed improved methods for DPoS [25,26,27]. Currently, it has been used in digital currency, supply chain management, the Internet of Things and other fields. For example, BitShares uses the DPoS consensus to implement a decentralized trading platform, and EOS uses it to implement a decentralized application platform. However, the DPoS consensus algorithm has several significant drawbacks: (i) The limitation of “one ballot, one vote”. Generally, in a simple network with a few nodes, it is easy for a node to choose one node as its own representative. However, in a more complex blockchain network, it is more reasonable for a node to choose multiple alternative nodes as its representatives. (ii) Low degree of decentralization and node voting enthusiasm. Generally, the more votes a node receives, the easier it is to be elected as an accounting node. Therefore, nodes may bribe other nodes so that rewards and the privileges of generating blocks are only obtained by a few nodes, which violates the principle of decentralization. In addition, most of the nodes are not rewarded for their participation in voting, which reduces the participation enthusiasm of nodes. (iii) Malicious nodes. In a complex network environment, it cannot be guaranteed that all nodes are normal nodes. When malicious nodes participate in voting or become accounting nodes, the security of the system will be seriously affected. Unfortunately, the DPoS algorithm lacks an effective mechanism for driving out malicious nodes, so malicious nodes can be active in the network for a long time. To address the above issues, this paper improves the DPoS consensus algorithm with the following main contributions:We propose a reputation value calculation method for nodes based on PageRank, which calculates the degree of trust that each node obtains from other voting nodes. In this way, a node can cast multiple votes, making the voting more fair and just.We propose a voting enthusiasm measurement method based on the GN algorithm, which calculates the voting enthusiasm of each node. Notably, the nodes that actively vote can receive corresponding rewards; thus, the enthusiasm of the nodes will be improved.We propose a credit incentive mechanism and a comprehensive election mechanism. First, the delegated nodes are comprehensively elected, which makes the selection of nodes more decentralized. Second, after consensus is completed, normal nodes will receive credit rewards, while malicious nodes will be added to the blacklist and removed from the consensus process, which will improve the security of the blockchain system.

## 2. Related Works

### 2.1. DPoS Improvement Studies

The DPoS algorithm is a combination of centralization and decentralization, and in theory, it enables each node to become a delegator through the voting election mechanism. Generally, if an accounting node violates the agreement not to do evil, its privilege of generating blocks will be cancelled, and the blockchain system will elect a new node as a delegator. Specifically, when the system has only a few nodes offline, multiple productions, a few forks, and network fragmentation, the DPoS consensus mechanism can guarantee correctness. However, when the voting nodes are bribed and the delegated nodes are manipulated, the degree of decentralization of the system is significantly reduced, and rewards and the privileges of generating blocks are only held by a few nodes. This will cause other nodes to give up voting because they cannot receive rewards, resulting in a decrease in voting enthusiasm. In addition, the consensus algorithm does not deal with malicious nodes quickly. As a result, when malicious nodes have the privileges of generating blocks and behave maliciously, the blockchain system will become insecure, unreliable and inefficient.

To address these problems, researchers have devised some improved schemes. For example, Wang et al. [28] designed a new dynamic voting mechanism to reduce the risk of voting attacks and introduced a Bayesian voting algorithm to calculate node scores and adjust voting weights according to voter objectivity. Wang et al. [29] adopted a verifiable multiparty random number generator (mRNG) to determine leaders, making the election of leaders unpredictable and publicly verifiable; thus, the fairness and validity of the election were greatly improved. Liu et al. [30] proposed a novel voting scheme using adjacency voting and a comprehensive selection of representative nodes by the average fuzziness of fuzzy values; thus, the security and fairness of the blockchain were greatly improved. Liu et al. [31] proposed the K-DPoS consensus algorithm to select reliable nodes from the delegator queue in advance by the K-means clustering algorithm, which solved the problem of node aggregation and malicious nodes. Wang et al. [32] proposed the CW-DPoS consensus algorithm, which used the k-shell algorithm to calculate the activity of the nodes and introduced credit value attributes to the nodes. It used a combination of node activity and weighted voting to select representative nodes, which improved the activity of the nodes and prevented malicious nodes from becoming accounting nodes.

As mentioned before, we compare and analyze the voting mechanism, the rewards and penalties for nodes, and the selection of representative nodes in related studies, as shown in Table 1. It can be seen from the table that the voting mechanism of most of the improved algorithms is “one ballot, one vote”. In addition, they mainly improve the selection of representative nodes and select more credible nodes, thereby reducing the probability of malicious nodes becoming accounting nodes. However, the voting mechanism of the above algorithms is obviously not reasonable enough to choose a node as its representative node in a complex network environment. In addition, the rewards and penalties for the nodes are not focused enough. In particular, it is difficult for nodes with low credit value to obtain the right to generate blocks and receive token rewards, which reduces the activity of the nodes. In conclusion, the current studies lack a comprehensive consideration of node reward allocation, election fairness, network security, and node enthusiasm. For this reason, this paper presents an improved delegated proof of stake algorithm based on community discovery (CD-DPoS), which can allow nodes to vote for multiple alternative nodes, deal with malicious nodes quickly, and effectively improve node enthusiasm and network security.

### 2.2. Preliminaries

#### 2.2.1. PageRank Algorithm

The PageRank algorithm, also known as the web ranking algorithm, is an iterative algorithm proposed by two Google founders in 1996. The algorithm comprehensively ranks web pages based on indicators such as importance and relevance, and the higher the ranking, the higher the probability of being visited. More specifically, for each web page (node), its importance (the probability of being visited) can be obtained iteratively by Formula (1).
(1)$$PR(j)=\frac{\left(1-d\right)}{N}+d\times \sum_{i\in B_{j}}\frac{PR(i)}{L(i)}$$
where *j* is the web page to be evaluated, and *N* and *d* denote the total number of pages and the damping factor, respectively. *B_j_* denotes the set of pages linked to page *j*, *PR*(*i*) denotes the PageRank value (*PR* value) of web page (node) *i*, and *L*(*i*) denotes the total number of other web pages linked to by page *i*. It should be noted that for any page *i* of *B_j_*, its effect on page *j* is the *PR* value of page *i* itself divided by *L*(*i*), which means that the *PR* value of page *i* is equally distributed to each of its outgoing links. The damping factor *d* indicates the probability of a user following a page skip link to another web page, while 1-*d* indicates the probability of going directly to the website. In addition, *d* solves the problem of rank leaks and rank sinks to some extent. A rank leak is defined as a web page without incoming links; such a web page absorbs the *PR* values of other web pages, which will eventually reduce the *PR* values of the other web pages to 0. A rank sink is defined as a web page with only outgoing links and no incoming links; as the process of calculation iterates, it will eventually reduce the *PR* value of the web page to 0.

#### 2.2.2. GN Algorithm

At present, there is no clear definition of the term community. Newman and Gievan gave a typical definition: a community is a subgraph of a graph that contains a denser number of nodes than the rest of the graph. In other words, a graph is said to contain a community if the number of links within a subgraph of the graph is higher than the number of links between subgraphs. In recent years, the GN algorithm proposed by Gievan and Newman has become a standard algorithm for community structure analysis. Generally, the algorithm starts from the whole network and continuously removes the edges with the largest edge betweenness from the network to obtain the best community structure. The edge betweenness is defined as the proportion of shortest paths in the network that pass through the edge, as shown in Figure 1. It shows the distribution of nodes, and the graph is searched with node 7 as the source. In particular, the edge betweenness of the edges closest to the leaf node is 1, such as for nodes 1, 2, 4 and 6. However, the edge betweenness of another edge is the sum of the edge betweennesses of the edges closest to it and 1. For instance, the edge betweenness between node 3 and node 7 is 3, which is equal to the edge betweennesses of the edges that depend on node 3 plus 1, more specifically, the edge betweennesses of the edges between node 3 and node 1 and 2. Since the graph is searched without passing through the edge between node 1 and node 2, the edge betweenness of this edge between node 1 and node 2 is 0.

## 3. Proposed CD-DPoS Algorithm

### 3.1. Algorithm Model

The proposed CD-DPoS consensus algorithm model is shown in Figure 2, which is divided into the following four steps:Each voting node starts voting, and each node can vote for multiple alternative nodes. Afterward, the *PR* value (reputation value) of each node is calculated, and it is calculated by a reputation calculation method based on PageRank, which will be discussed in detail in Section 3.3. In short, the node’s reputation value represents its level of trust, measured by the number of votes cast for it.The node voting graph formed by the votes is divided into more suitable communities by the GN algorithm. Finally, the *Q* value of each community, that is, the voting enthusiasm of the nodes, can be obtained and assigned to the corresponding node.The representative nodes are selected to generate blocks by comprehensively considering each node’s voting enthusiasm and the reputation value.After a round of consensus is completed, each node’s credit value is computed through the credit incentive mechanism, which is measured by the node’s behavior. Specifically, when an accounting node successfully generates a block, it and the nodes that voted for it are rewarded with the corresponding credit value. In contrast, when an accounting node maliciously generates a block, it and the nodes that voted for it have the corresponding credit value deducted as a penalty.

### 3.2. Block Structure and Node Information

The block structure and node information of the blockchain network proposed in this article are shown in Figure 3. It can be seen that Figure 3a shows the block structure in the blockchain, which contains information such as the version of the block header, previous hash, Merkle root, proof of block generation, timestamp, and block height. Figure 3b shows the information of each node, including the port address *p*, comprehensive reputation value *r*, credit value *c*, modularity metric value *Q*, degree of the node *d*, and number of votes *v*. Specially, this information is used to comprehensively elect representative nodes, where *p* denotes the unique identifier of the node identity; *r* denotes the combined value of the node’s reputation and voting enthusiasm, which is utilized to select the representative node; and *c* denotes the credibility of the node, and the value is changed according to the behavior of the node after each round of consensus is completed. *Q* denotes the node’s voting enthusiasm, and the higher the node’s enthusiasm, the larger the corresponding *Q* value; *d* denotes the degree of the node after the graph is formed by the voting relationship; and *v* denotes the number of votes cast by the node during the voting process, which determines the degree of reward or punishment that the voting node should receive at the end. In particular, when the voting node votes for the representative node that successfully generates a block, the smaller *v* is, the greater the reward it will receive. For example, if two nodes, *x* and *y,* vote for an accounting node, and node *x* casts out two votes while node *y* casts out three votes, then if the accounting node generates a block honestly, node *x* receives a greater reward. Conversely, when a voting node votes for an accounting node that maliciously generates a block, then the smaller *v* is, the higher the penalty the node receives.

### 3.3. Reputation Value Calculation for Nodes Based on PageRank

First, each node has a credit value attribute, represented by *c*, and its initial value is 70. When a node votes for multiple alternative nodes during the voting process, its trust in multiple candidate nodes is equal, so the reputation value calculation method for the nodes based on PageRank is proposed. It will divide the *c* of the voting node among other nodes equally according to *v*. A simple example is shown in Figure 4. Node 1 has a credit value of *c* and voted for the other three nodes, where *v* is 3. Therefore, Node 2, Node 3, and Node 4 will each receive a credit value of *c*/3. After voting is finished, the reputation value (represented as the *PR* value) of the nodes is calculated by the following formula:
(2)$$PR_{x}=\sum_{i=1}^{N}\frac{C_{i}}{V_{i}}$$
where the reputation value of node *x* is denoted by *PR_x_*, which represents the degree of trust of the node; the credit value of node *i* is denoted by *C_i_*; the number of votes cast by node *i* is denoted by *V_i_*; and the number of nodes that voted for node *x* among all voting nodes is denoted by *N*. The *PR* value calculation process for the nodes is shown in detail in Algorithm 1.

**Algorithm 1** Reputation value calculation algorithm for nodes based on PageRank
**Input:**
* nodes*

**Output:**
* PR_x_*
 1:   **for** i=0→n−1 **in** *nodes* **do**      //Traverse the nodes in the network 2:        **if** *node_i_*.getCredit()<60:           //Nodes are malicious nodes 3:              *nodes* = *nodes* − *node_i_*          //Eliminate malicious nodes 4:              **continue**                           //Go to the next cycle 5:  
          **end if**
 6:            *reputation_x_* = *credit_i_*/*vote_i_*   //Trust given by node *i* to node *x* 7:            *PR_x_* = *PR_x_* + *reputation_x_*     //Calculate the reputation value obtained by node*x* from other nodes 8:  
**end for**
 9:
**return** *PR_x_*

### 3.4. Node Enthusiasm Measurement Based on the GN Algorithm

After voting is finished, the voting relationship between nodes is represented by connected line segments, so a graph can be formed after voting is completed. Afterward, the enthusiasm of each node is calculated by the GN algorithm through graph theory. In the GN algorithm, there is a modularity metric value *Q*, which is an indicator of the strength of the community structure. It is calculated based on the line segments in the graph, which determines the stability of the structure. In particular, the higher the *Q* value is, the closer the relationship between nodes, which means that the voting enthusiasm of the nodes in the community is higher. The *Q* value is calculated by the following formula:(3)$$Q=\frac{1}{2}\sum_{\mathrm{vw}}\left[A_{vw}-\frac{k_{v}k_{w}}{2m}\right]\sigma (c_{v},c_{w})$$
where *A_vw_* is an element of the adjacency matrix of the graph. If node *v* is connected to node *w*, the value of *A_vw_* is 1; otherwise, it is 0. *k_v_* and *k_w_* denote the degrees of node *v* and *w*, respectively. If nodes *v* and *w* belong to the same community, the value of the *σ*(*c_v_*, *c_w_*) function is 1; otherwise, it is 0. In addition, the total number of edges in the graph is denoted by *m*.

An example of the GN algorithm is shown in Figure 5. There are 14 nodes in the figure, and the voting relationships between nodes are represented by connected line segments. The algorithm mainly consists of the following steps:Calculate the edge betweenness of each edge in the graph, as shown in Figure 5a.Delete the edges with the largest edge betweenness in the graph. For example, the largest edge betweenness in Figure 5a is 3, and the edges with this value are those between node 7 and node 3, between node 7 and node 5, between node 8 and node 9, and between node 8 and node 12. After deleting them, we obtain a new topology, as shown in Figure 5b. Then, calculate the modularity metric value *Q* of the corresponding community and the sum *Q_n_* of the *Q* value of each community.Repeat steps 1 and 2 until all edges in the graph are removed, as shown in Figure 5c.Find the maximum value *Q_n_*, and label the corresponding community structure to form a relatively stable community, as shown in Figure 5d. The groups of nodes 1, 2, and 3; nodes 4, 5 and 6; nodes 9, 10 and 11; and nodes 12, 13 and 14 all form small communities. However, the groups formed by nodes 1, 2, 3, 4, 5, 6, and 7 and by nodes 8, 9, 10, 11, 12, 13 and 14 form larger communities.

The node enthusiasm measurement based on the GN algorithm is shown in Algorithm 2.
**Algorithm 2** Node enthusiasm measurement based on the GN algorithm**Input:** *G***Output:** *Q, community_max_* 1:  **while** len(*G*.edges()) > 0 **do**  //Determine whether any node in the graph is a community 2:        *edge* = max(edge_betweenness(*G*))    //Find the edge with the largest edge betweenness in the graph 3:        *G*.remove_edge(edge[0], edge[1])      //Remove the edges with the largest edge betweenness 4:        *components* = connected_components(*G*)    //The connected subgraph is obtained after removing the edges 5:        **if** len(*components*)! = len(*nodes*)**:**      //Determine whether any node of the connected subgraph is a community 6:               *Q*, *community* = cal_Q(*components*, *G*) //Calculate the modularity metric value *Q* 7:               **if** *Q_n_* > *Q_max_***:** //Determine whether the sum *Q_n_* of *Q* values is maximum 8:                         *Q*, *Q_max_*, *community_max_* = *Q*, *Q_n_*, *community* //Record the communities that are now formed, *Q* and *Q_n_* 9:  
             **end if**
 10:  
      **end if**
 11:  **end while** 12:
**return** *Q*, *community_max_*

### 3.5. Comprehensive Election Mechanism

To make the selection of representative nodes fairer and more trustworthy, the reputation values of the nodes are obtained through Algorithm 1 as described in Section 3.3, and the enthusiasm of the nodes are obtained through Algorithm 2 as described in Section 3.4. Afterward, we calculated the comprehensive reputation values of the nodes with the following formula:(4)$$I_{x}=\eta \cdot (Q(x)\cdot k_{x})+\phi \cdot 1\%\cdot \sum_{i=1}^{N}\frac{C_{i}}{V_{i}}$$
where *I_x_* denotes the comprehensive reputation value of node *x*, *Q*(*x*) denotes the modularity metric value of node *x*, *k_x_* denotes the degree of node *x*, *N* denotes the number of nodes that voted for node *x* among all voting nodes, *C_i_* denotes the credit value of node *i*, *V_i_* denotes the number of votes cast by node *i*, and *η* and ϕ denote the weight factors; their values will be discussed in detail below. After the calculation is completed, we sort the comprehensive reputation values of the nodes from high to low and select the highest node as the accounting node.

### 3.6. Credit Incentive Mechanisms

For the convenience of calculation, we set c∈[0,100] and divide them into three different levels as follows: (i) Nodes with c∈[90,100] are marked as good nodes. A good node is defined as a node that has participated in the process of consensus for a long time without performing malicious actions; that is, it is an honest node. (ii) Nodes with c∈[90,100] are marked as ordinary nodes. An ordinary node is defined as a node that has recently joined the blockchain to participate in the process of consensus, that has voted one or two times for accounting nodes that have failed to generate blocks, or that has failed to generate blocks one or two times after becoming an accounting node. (iii) Nodes with c∈[0,60) are marked as malicious nodes. A malicious node is defined as having voted several times in the consensus for an accounting node that has failed to generate blocks honestly or has maliciously generated blocks multiple times after becoming an accounting node.

It should be noted that nodes that join the blockchain network are required to pay a certain deposit. In this way, if a node is malicious, the system will mark it, prevent it from participating in the process of consensus again, and deduct its deposit. Therefore, when a node aims to act maliciously, it will comprehensively consider the benefits of its own behavior, which reduces the possibility of the node being malicious to some extent. Generally, the credit value of a node that has just joined the blockchain is set to 70, and the node needs to actively vote or honestly generate blocks in the subsequent consensus process to increase its credit value. There is no doubt that if the credit value of a node is higher, then its credibility in the blockchain network is higher, which will lead to more trust (larger number of votes) and a higher probability of running as an accounting node.

After each round of consensus, the system gets the initial credit value of the node, and if the accounting node successfully generates a block, both it and the nodes that voted for it will increase their credit values according to the corresponding reward formula. On the contrary, if the accounting node fails to generate a block, both it and the nodes that voted for it will be penalized with a penalty formula that deducts their credit values, as shown in Figure 6. It shows the calculation rules for updating the node credit value, which is used for an individual formula to reward or penalize the node as an incentive mechanism for the node. Here, $c_{0}^{g}$ and $c_{0}^{n}$ denote the initial credit values of good nodes and ordinary nodes, respectively, and $c_{1}^{g}$ and $c_{1}^{n}$ denote the credit values of good nodes and ordinary nodes after the consensus is completed, respectively. To implement credit rewards and punishments, six parameter factors (λ,α,β,γ,ξ and *θ*) are set, where α>λ>β>1>γ>ξ>θ>0.

For an accounting node with a *c* reaching 90, the largest credit value will be deducted for one malicious deed, as shown below:(5)$$c_{1}^{g}=c_{0}^{g}-\alpha$$when an accounting node generates a block honestly, its credit value will be increased by *Υ*, as shown below:(6)$$c_{1}^{g}=c_{0}^{g}+\gamma$$

For an accounting node with c∈[60,90), its credit value will be reduced by *λ* for one malicious deed, as shown below:(7)$$c_{1}^{n}=c_{0}^{n}-\lambda$$when an accounting node successfully generates a block, its credit value will be increased by θ×α, as shown below:(8)$$c_{1}^{n}=c_{0}^{n}+\theta \times \alpha$$

For nodes with c∈[0,60), the system will set *c* = 0, prevent them from participating in the process of consensus again, and it will deduct their security deposit as the cost of their bad behavior. In addition, to increase the enthusiasm of node voting, the system rewards nodes that actively vote with a certain credit value. According to different situations of voting nodes, the system will use different incentives as follows.

When the votes cast by a good node are not for the node chosen as the honest accounting node, the reward for the good node can be computed by the following formula:(9)$$c_{1}^{g}=c_{0}^{g}+\theta \times \gamma \times \xi$$when the votes cast by a good node are for the node chosen as the honest accounting node, the system will reward the good node according to its voting status. The reward can be computed by the following formula:(10)$$c_{1}^{g}=c_{0}^{g}+\theta \times \beta \times \gamma$$

When the votes cast by an ordinary node are not for the node chosen as the honest accounting node, the reward for the ordinary node can be computed by the following formula:(11)$$c_{1}^{n}=c_{0}^{n}+\theta \times \beta$$when the votes cast by an ordinary node are for the node chosen as the honest accounting node, the system will reward the ordinary node according to its voting status. The reward can be computed by the following formula:(12)$$c_{1}^{n}=c_{0}^{n}+\beta \times \xi$$

In contrast, for voting nodes that vote for malicious nodes, the system will deduct a certain credit value as a penalty according to the situation; when a good node votes for an accounting node that maliciously generates a block, its punishment can be computed by the following formula:(13)$$c_{1}^{g}=c_{0}^{g}-\alpha \times \gamma$$when an ordinary node votes for an accounting node that maliciously generates a block, its punishment can be computed by the following formula:(14)$$c_{1}^{n}=c_{0}^{n}-\beta \times \gamma$$Therefore, nodes can only continuously increase their credit values by actively voting and generating blocks honestly, thereby increasing their probability of becoming accounting nodes.

## 4. Theoretical Analysis of Algorithm Performance

### 4.1. The Degree of Decentralization

Generally, to prevent malicious nodes from becoming accounting nodes, existing improvement schemes usually introduce credit mechanisms. Typically, as described by Hu et al. [27], nodes with high credit values can become accounting nodes. That is, it is difficult for newly added nodes or ordinary nodes with low credit values to obtain accounting opportunities, and its degree of decentralization is not high. In our method, as described in Equation (4), *Q*(*x*) takes values in the range [−1, 1], and *C_i_* takes values in the range [0, 100]. Suppose a node receives 10 votes; then, *k_i_* is also 10, and the node’s *V_i_* takes an integer value in the range [1, 3]. Obviously, the value of $1\%\cdot \sum_{i=1}^{N}\frac{C_{i}}{V_{i}}$ is within [1, 10], and the value of $Q(x)\cdot k_{x}$ is within [−10, 10]. Specifically, assuming that an ordinary node *a* (that newly joined the network) actively votes and a good node *b* (that has participated in consensus for a long time) is far less active than node *a*, the weighting factors *η* and ϕ are adjusted appropriately so that the comprehensive reputation value *r* of node a is greater than the *r* of node *b*, where η+ϕ=1; thus, node *a* obtains an accounting opportunity. As mentioned before, the newly joined nodes that actively vote can still become accounting nodes, which improves the voting enthusiasm of the nodes and the degree of decentralization to some extent.

### 4.2. System Security

After each round of consensus, the credit values of all nodes are updated by the credit incentive mechanism proposed in this paper. In particular, if a malicious node becomes an accounting node and maliciously generates blocks, both the accounting node itself and the nodes that voted for it will be punished. It is worth mentioning that the penalty received by the voting nodes are related to the number of votes they cast. For example, if the voting node votes less, it indicates that the relationship between the voting node and the accounting node is closer, so the penalty is heavier; that is, the deducted credit value is greater. Conversely, if the voting node votes more, the penalty will be lighter; that is, the deducted credit value is smaller. In particular, when a node’s credit value falls below the threshold, the system will judge it as a malicious node. Then, it will be blacklisted, banned from participating in the process of consensus again, and its security deposit will be deducted. Through this approach, the ability to evict malicious nodes from the blockchain network is effectively improved, thereby enhancing system security.

## 5. Experimental Results and Analysis

The experimental analysis of the CD-DPoS algorithm is divided into four parts to verify its correctness and effectiveness: the credit incentive mechanism, the selection of accounting nodes, system security, and algorithm performance. Specifically, this experiment used an Intel i5-6200U processor, an 8 GB memory environment, and PyCharm 2020.1, and the experimental simulations were conducted using Python 3.8.

### 5.1. Credit Incentive Mechanism

According to the criteria in Deng et al. [33] for determining the malicious behavior of nodes, we quantified the behavior of malicious nodes and judged a node as malicious when it had performed three malicious actions. Therefore, we made appropriate adjustments to the parameter factors λ,α,β,γ,ξ and *θ*, where *α* took the value 22.5, *Υ* took the value 0.3, *λ* took the value 10, and θ×α took the value 3. In addition, the reward and punishment methods were adjusted by the following formulas:(15)$$\theta \times \gamma \times \xi =1\%\times V_{i}$$
(16)$$\theta \times \beta \times \gamma =10\%\cdot \left[{log}_{2}^{\frac{1}{V_{i}}+1}-0.2\right]$$
(17)$$\theta \times \beta =\theta_{0}+10\%\times V_{i}$$
(18)$$\beta \times \xi =30\%\times \frac{1}{V_{i}}$$
(19)$$\alpha \times \gamma =15\times \frac{3}{V_{i}}$$
(20)$$\beta \times \gamma =5\times \frac{3}{V_{i}}$$
where *V_i_* denotes the total number of votes cast by node *i*. When the number of times a node acts maliciously reaches 3, the node’s credit value will be lower than the threshold of 60, and its credit value will be set to 0. The reward and punishment results for the nodes’ credit values are shown in Figure 7. In particular, node 1 and node 2 are ordinary nodes, and node 3, node 4 and node 5 are good nodes. As can be seen, the credit values of node 1 and node 3 grow steadily. Conversely, node 2 and node 5 have 3 malicious behaviors, reducing their corresponding credit values below the threshold; finally, the credit values are set to 0. Specially, node 4 has one malicious behavior, reducing the credit value once.

In the diagram, the ordinary node’s (node 1) credit value grows faster than that of the good node (node 3), so the ordinary nodes can be encouraged to vote actively and generate blocks honestly. It should be noted that although node 2 voted for malicious nodes twice, the credit value deducted in these two times is different because the total number of votes cast in these two times is different. Specifically, the total number of votes cast in the first vote is less than that in the second vote; because node 2 is closer to the malicious node, its penalty is heavier, so the score deducted for the first vote is greater than that deducted for the second vote. In addition, it can be seen from Figure 7 that when node 5 acts maliciously three times, its credit value will be lower than the threshold and will be reset to 0. However, node 4 acted maliciously once, but it can participate in the process of consensus again. Therefore, this algorithm can effectively identify malicious nodes.

### 5.2. Accounting Node Selection

To demonstrate the effectiveness of the accounting node selection mechanism, we provide an example of the comprehensive reputation value change of the nodes, as shown in Figure 8, where node 1 and node 2 are ordinary nodes, and nodes 3, 4 and 5 are good nodes. It can be seen that the comprehensive reputation values of node 1 and node 3 grow steadily. Additionally, node 2 and node 5 experience a decrease in their comprehensive reputation value three times due to three instances of malicious behaviors, and finally, their comprehensive reputation values are 0. However, node 4 has one malicious behavior, and its comprehensive reputation value also decreases once accordingly, but it continues to grow slowly.

In addition, as shown in Figure 8, when an ordinary node (node 1) actively votes and honestly generates blocks, its comprehensive credit value can exceed that of a good node (node 3) after a certain consensus round. This is because although the degree of trust received by node 1 from other nodes in the voting process is lower than that of node 3, the enthusiasm of node 3 for participating in voting is far less than that of node 1. In particular, node 4 made a mistake once and can continue to participate in the process of consensus afterward, so its comprehensive credit value can continue to grow after decreasing once. In contrast, node 2 and node 5 have made three mistakes and are judged as malicious nodes, and the system marks them and sets their credit values to 0, prohibiting them from joining the consensus again. To summarize, when nodes actively vote and generate blocks honestly, the newly joined nodes can still obtain accounting opportunities, which means that the privilege of generating blocks will not always be in the hands of the node with the highest credit value. Therefore, this makes the blockchain more decentralized.

### 5.3. Security

In this experiment, we compared the probability of malicious nodes becoming accounting nodes by three methods, namely, our CD-DPoS algorithm, the CW-DPoS algorithm of Wang et al. [32], and the DPoS consensus algorithm. Figure 9 shows the experimental comparison results. It should be noted that as the number of consensus rounds increases, the probability of a malicious node becoming an accounting node is approximately 60% using the DPoS algorithm, while the probability of a malicious node becoming an accounting node decreases gradually using the CW-DPoS and CD-DPoS algorithms.

Due to the lack of corresponding eviction processing for malicious nodes in the DPoS algorithm, the probability of malicious nodes becoming accounting nodes is high, and the system’s security is poor. Conversely, the CW-DPoS algorithm selects trusted representative nodes by introducing credit values. Thus, as the number of consensus rounds increases, the probability of malicious nodes becoming accounting nodes gradually decreases until it reaches 0. In addition, our CD-DPoS algorithm adopts a relatively reasonable reward and punishment mechanism to reward and punish nodes. In particular, the nodes that vote for malicious nodes will also be punished, which enables malicious nodes to be detected and eliminated quickly, greatly improving the quality of the network nodes. The experimental results show that our CD-DPoS algorithm improves the quality of nodes and makes the blockchain system more secure and reliable than the CW-DPoS algorithm.

### 5.4. Algorithm Performance

When the system transactions reach hundreds of thousands, we compared the latency and throughput of DPoS, CW-DPoS, and CD-DPoS, and the experimental results are shown in Figure 10 and Figure 11.

Figure 10 shows that as the number of consensus nodes increases, the consensus latency of all three algorithms shows an upward trend. In particular, the latency of the DPoS algorithm slowly increases because the DPoS algorithm can be applied to large networks and has lower energy consumption. However, the CW-DPoS and CD-DPoS algorithms have a larger latency growth trend when the number of consensus nodes is less than 150. This is because the addition of new nodes leads to an exponential increase in the number of line segments of the graph formed by the voting relationship between voting nodes; thus, it significantly increases the time required for consensus. It should be noted that when the relationship between the nodes stabilizes and nodes are added again, the latency of both algorithms shows a slow growth trend. Specifically, compared to CW-DPoS, the CD-DPoS algorithm takes more time to complete because all voting nodes need to be rewarded and punished to improve the enthusiasm of the nodes and eliminate more malicious nodes in the credit incentive mechanism. However, the extra time taken by CD-DPoS is within the acceptable range, in contrast to that of CW-DPoS.

In addition, Figure 11 shows that as the number of nodes increases, the throughput of the DPoS, CW-DPoS and CD-DPoS algorithms all show a decreasing trend because the system takes more time to complete the consensus process. In particular, the transaction throughput is highest when using the DPoS algorithm. As for the other two algorithms, the throughput of CD-DPoS is higher than that of CW-DPoS when the number of nodes is greater than 150. This is because the CW-DPoS algorithm iterates over all the nodes in the graph continuously, and it is necessary to re-traverse the nodes every time a new node is added. Significantly, the community structure is not easy to change once it is stabilized, so the time for CD-DPoS to complete the consensus when adding new nodes is lower than that of CW-DPoS. Therefore, the throughput of CD-DPoS is higher than that of CW-DPoS for a certain number of nodes. The experimental analysis shows that the more nodes there are in a network with more than 150 nodes, the more obvious the advantage of CD-DPoS over the CW-DPoS algorithm.

## 6. Conclusions

DPoS, as one of the most mainstream consensus algorithms for blockchains, has received widespread attention from researchers recently. However, DPoS and its typical improvement algorithms still have problems, such as “one ballot, one vote”, a low degree of decentralization, and malicious nodes. To address these issues, we propose a novel DPoS algorithm based on community discovery and the credit incentive mechanism, called CD-DPoS, which can allow nodes to vote for multiple alternative nodes, deal with malicious nodes quickly, reasonably distribute node rewards, and effectively improve node enthusiasm and network security. The experimental simulation results show that the CD-DPoS algorithm proposed in this paper has a higher degree of decentralization, can expel malicious nodes in a more timely manner, and has a stable throughput at a larger network node size. Nevertheless, the credit incentive formulas in this paper still have shortcomings, and the speed of node consensus also needs to be improved. In addition, many members will not vote, regardless of how easy you make it for them to vote in the blockchain ecosystem. In the future, we will strategically require all nodes to vote, so that nodes can get their benefits in active and correct voting activities. Additionally, a non-chain or master–slave multi-chain storage structure and a structured communication topology will be used for communication to further improve the DPoS mechanism.

## Figures and Tables

**Figure 1 entropy-25-01320-f001:**
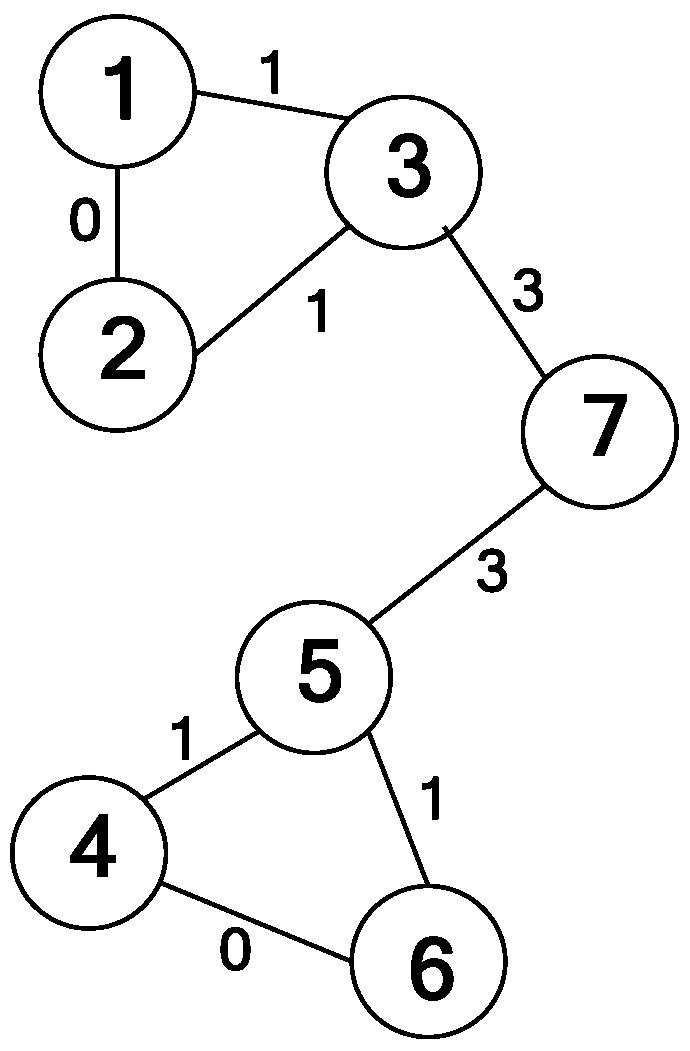
Definition of edge betweenness.

**Figure 2 entropy-25-01320-f002:**
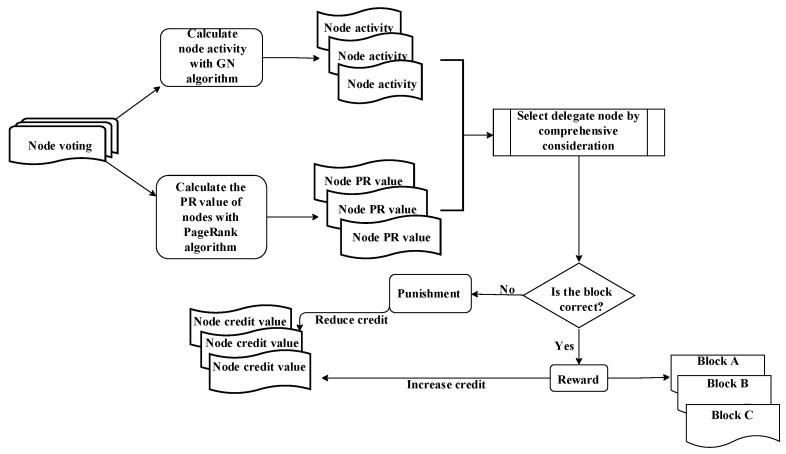
CD-DPoS consensus algorithm model.

**Figure 3 entropy-25-01320-f003:**
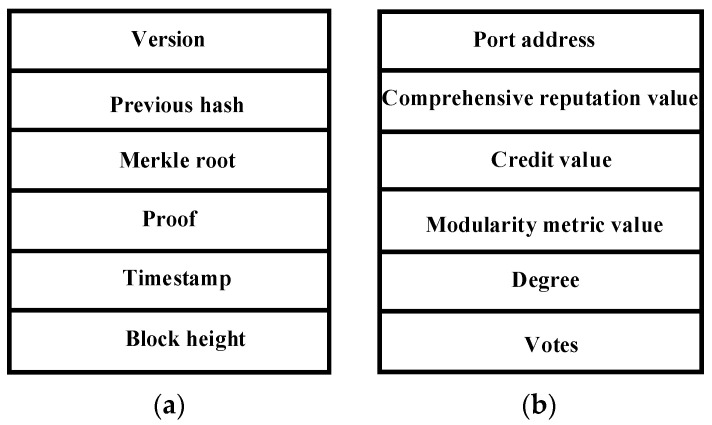
Block structure and node information. (**a**) Block structure; (**b**) node information.

**Figure 4 entropy-25-01320-f004:**
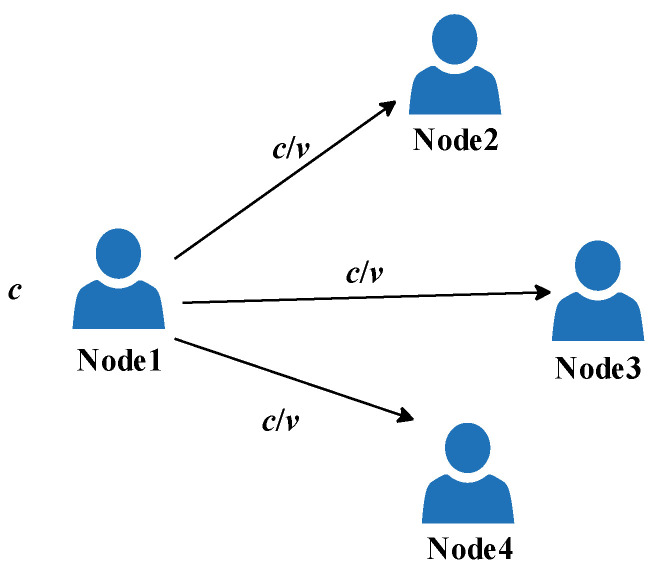
Allocation of credit values to nodes.

**Figure 5 entropy-25-01320-f005:**
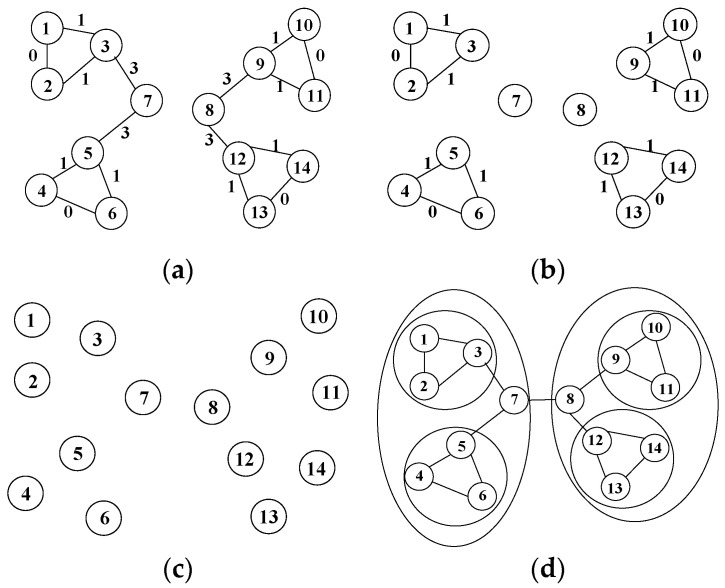
GN algorithm example. (**a**) Calculate the edge betweenness of each edge; (**b**) delete the edges with the largest edge betweenness; (**c**) all edges are removed; (**d**) find the most suitable community.

**Figure 6 entropy-25-01320-f006:**
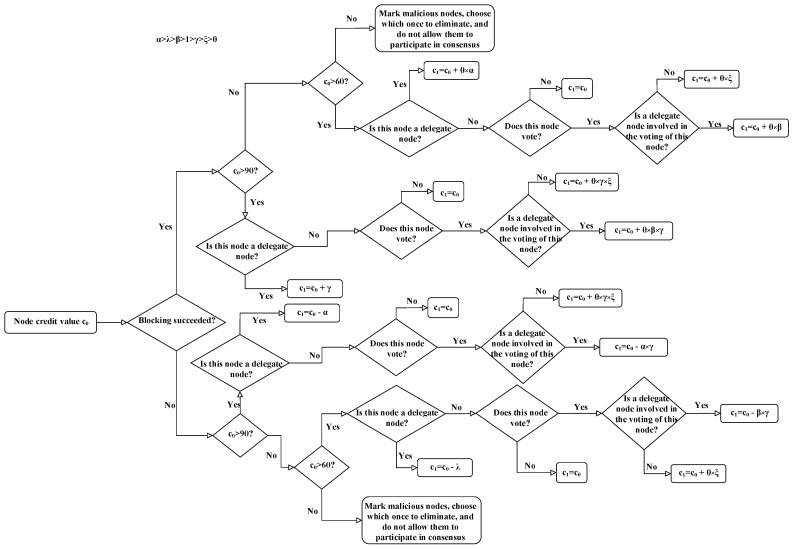
Calculation rules for updating node credit value.

**Figure 7 entropy-25-01320-f007:**
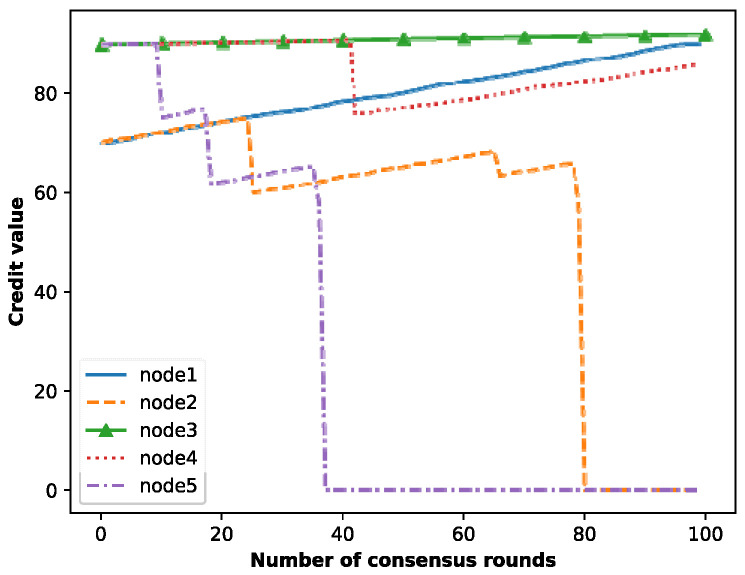
Credit value reward and punishment of nodes.

**Figure 8 entropy-25-01320-f008:**
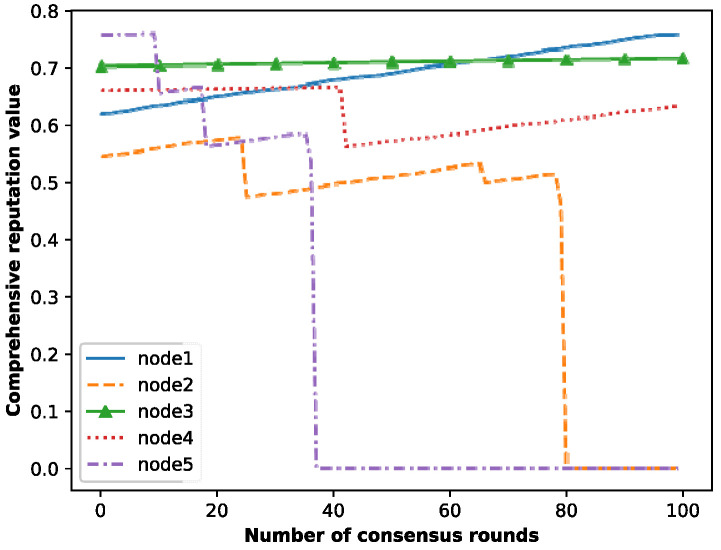
Comprehensive reputation values of nodes.

**Figure 9 entropy-25-01320-f009:**
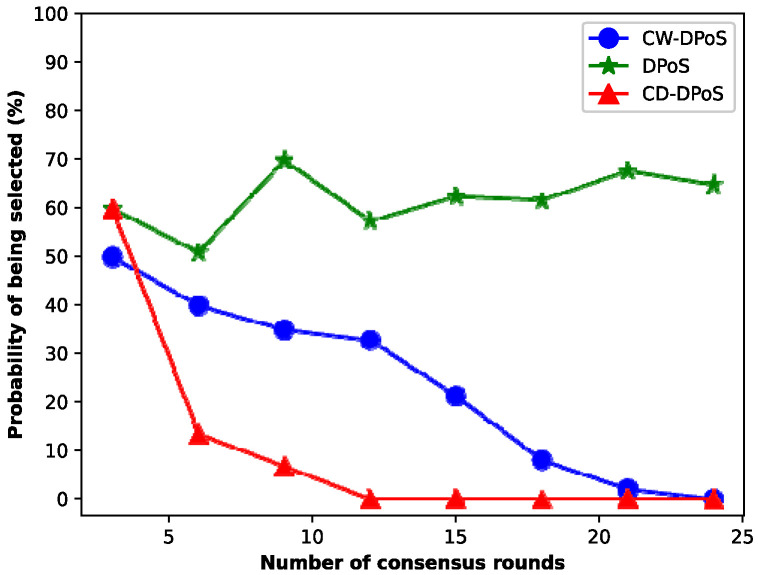
Probability of a malicious node becoming an accounting node.

**Figure 10 entropy-25-01320-f010:**
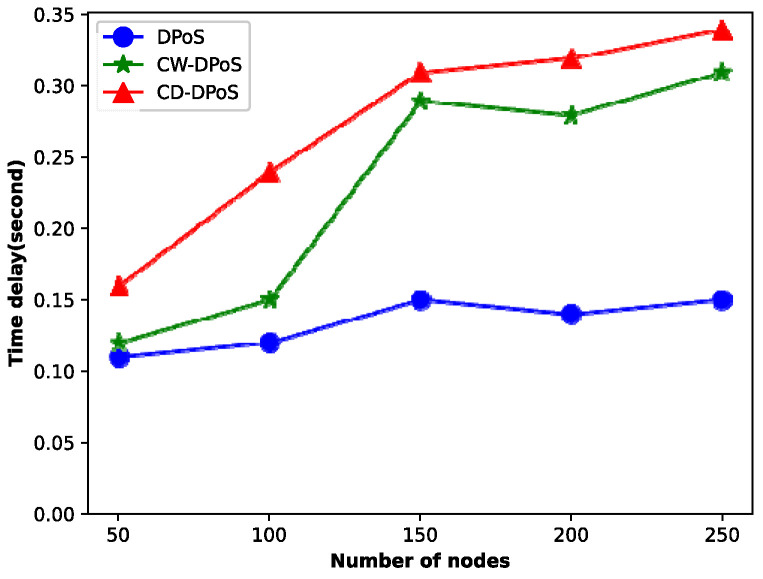
Latency comparison.

**Figure 11 entropy-25-01320-f011:**
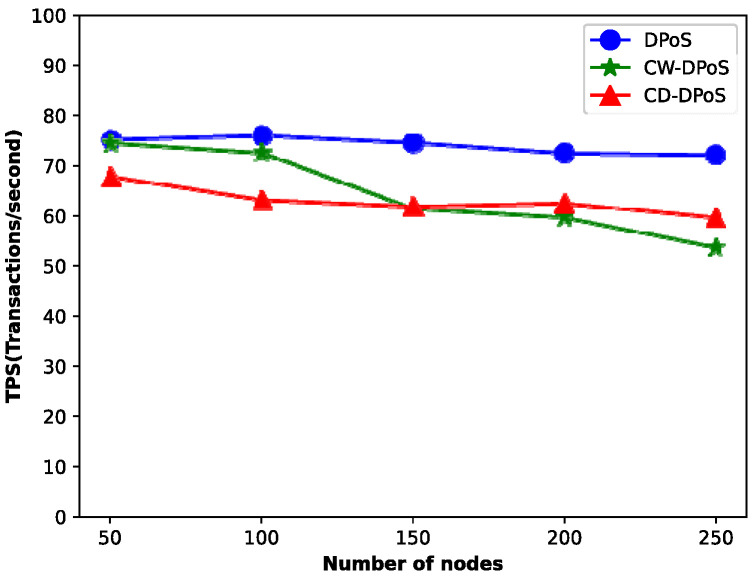
Throughput comparison.

**Table 1 entropy-25-01320-t001:** Comparative analysis of related works.

Algorithm	Voting Mechanism	Rewards and Penalties for Nodes	The Selection of Representative Nodes
Literature [28]	One ballot, one vote	High-risk nodes are removed	Nodes with high reputation value become accounting nodes
Literature [29]	One ballot, one vote	If the leader node is evil, it will be randomly replaced by other nodes	Random selection of nodes
Literature [30]	Multiple ballots, multiple votes	No presentation	Nodes with high reputation value become accounting nodes
Literature [31]	One ballot, one vote	No presentation	High-quality nodes become accounting nodes
Literature [32]	One ballot, one vote	Reward and penalize voting nodes and accounting nodes to the same extent	Comprehensive selection of accounting nodes based on node credit value and vote counts

## Data Availability

Not applicable.

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
