# Peer review of "Delegated Proof of Stake Consensus Mechanism Based on Community Discovery and Credit Incentive"

_entropy, 2023, doi:10.3390/e25091320_

Round 1
Reviewer 1 Report
The primary purpose of user reputation systems is to establish trust between unknown parties. Based on a reputation model, a reputation system enables the collection, aggregation and distribution of data about an entity, which can, in turn, be used to identify and predict the future actions of that entity. Using this data, e-commerce users can decide whom they will trust and to what degree. Reputation systems increase or decrease the user's ranking according to information collected about the user. They can, therefore, give a positive score, leaving the user with a better ranking, or they can give a negative reputation to punish dishonest behaviour. Thus, many online marketplace platforms have developed user reputation management systems that allow trading parties to submit a rating of the counterparty performed in a specific transaction, which will be available to all site visitors. A positive rating of my trading partner will likely increase trust in the counterparty's performance.
The paper presents a DPoS algorithm based on community discovery is designed, called CD-DPoS.
Strengths:
The presentation was clear and very understandable.
The paper generally presents appropriate organisation, and the points are advanced logically.
This paper provides an overview of the development of the area.
The system proposed is not only prominent but also very well written.
The Documentation of sources and references list are appropriate and up-to-date.
The figures and tables are suitable.
Weaknesses
The authors, apart from the limitations of the proposed mechanism, do not specify future work.
The system should be validated in a real system where hundreds of thousands of transactions occur.
The language needs revision of minor typos.
Author Response
We discuss further work at the end of the conclusion.
We tested the performance of the system over hundreds of thousands of transactions, the test results can be find in Section 5.4.
Reviewer 2 Report
Research: DPoS consensus mechanism based on community discovery 2 and credit incentive
Date: 04/08/23
Journal: Entropy, MDPI
This research is interesting. Having followed the journey of blockchain almost from the start I find this research potentially useful but some improvements are still needed.
Some improvements need to be made in terms of writing and presentation, formatting and citations.
The end of the introduction needs to be organized a little better, too many bullets.
Check that you use brackets consistently.
The end of page 17 needs to be organized better. There are similar issues elsewhere. Align the lines correctly please.
Many sentences are not wrong but could be nicer. For example: ‘is considered a disruptive innovation in computing after mainframes, personal computers and the internet’ would be better if it was like this for example ‘is considered one of the most disruptive innovations in computing after mainframes, personal computers and the internet’. The paper should be read again and opportunities to improve sentences should be utilized. If the papers flows better more people will read it.
I suggest putting blockchain as one of the keywords. This paper may be useful to people that would not think to search with the more specialized terms.
The issue about how decentralized blockchain is, and whether this is important to people is not the main point of this paper but it should be covered a little more thoroughly as it may influence someones decision to use this method. These sources should be used to better illustrate some of the issues fro the human perspective:
https://doi.org/10.1016/j.techsoc.2021.101745
http://link.springer.com/chapter/10.1007/978-3-319-11460-6_21#
https://ro.uow.edu.au/acis2016/papers/1/60/
The aim and research questions are clearly defined and aligned.
The methodology is suitable and seems to be implemented well.
For this research topic the originality is sufficient and suitable for this journal.
I hope the value in the guidance is clear.
I strongly encourage the authors to complete all the recommendations unless they have a very strong and convincing reason not to.
I wish the authors every success in their future endeavours.
The language needs some improvements, but I believe you can do it.
Author Response
We revise the formatting of the paragraph at the end of the Introduction.
We have carefully checked the brackets used in this paper.
We organized some sentences in the end and aligned the lines of the Algorithm 1.
We revised some sentences in the paper appropriately.
We add the word "blockchain" as a keyword.
We added reason for decentralization of blockchain in the Introduction section. Besides, the benefits of blockchain in transactions are also added.
Reviewer 3 Report
This article presents a new DPoS algorithm together with a simulation of its performance. I am personally not an expert in the mathematical part, so I cannot judge that section of the paper, but I have detected several issues which affect the contribution of the paper, and which I hope the authors can tackle:
1- The introduction of the paper could be more focused on blockchain governance issues. Why is DPoS important, and in which situations is used?
Some papers which could help authors frame the introduction about governance:
Pelt, R. V., Jansen, S., Baars, D., & Overbeek, S. (2021). Defining blockchain governance: a framework for analysis and comparison. Information Systems Management, 38(1), 21-41. Cowen, N. (2020). Markets for rules: The promise and peril of blockchain distributed governance. Journal of Entrepreneurship and Public Policy, 9(2), 213-226. Alston, E. (2020). Constitutions and blockchains: Competitive governance of fundamental rule sets. Case W. Res. JL Tech. & Internet, 11, 131. Lumineau, F., Wang, W., & Schilke, O. (2021). Blockchain governance—A new way of organizing collaborations?. Organization Science, 32(2), 500-521. van Haaren-van Duijn, B., Bonnín Roca, J., Chen, A., Romme, A. G. L., & Weggeman, M. (2022). The dynamics of governing enterprise blockchain ecosystems. Administrative Sciences, 12(3), 86.
2- The authors are not the only ones who have developed alternative consensus mechanisms, there's a plethora of similar work in the literature. This paper's literature review could emphasize a bit more the difference between the authors' work and similar work, by making a table comparing it to other papers such as the ones they cite plus... A quick Google search gave me many examples of other scholars working on DPoS consensus mechanisms
Bachani, V., & Bhattacharjya, A. (2022). Preferential Delegated Proof of Stake (PDPoS)—Modified DPoS with Two Layers towards Scalability and Higher TPS. Symmetry, 15(1), 4. Hu, Q., Yan, B., Han, Y., & Yu, J. (2021). An improved delegated proof of stake consensus algorithm. Procedia Computer Science, 187, 341-346. Chen, R., Wang, L., & Zhu, R. (2022). Improvement of Delegated Proof of Stake Consensus Mechanism Based on Vague Set and Node Impact Factor. Entropy, 24(8), 1013.
3- You propose three contributions of your method. In Figure 6, where you bring everything together, could you please highlight which parts of the mechanism provide each of the three benefits? That way, I think it will be easier for readers to understand the causal relationships of your mechanism. Also, I would suggest making Figure 6 bigger. Right now it's impossible to read if you print the paper.
4- In the conclusion, you should acknowledge the potential limitations of your model, as well as propose avenues for future research. How do you think for instance the system may behave when you change computers for people? My experience dealing with blockchain ecosystems is that many members won't vote regardless of how easy you make it for them to vote. It's a matter of human laziness, lack of commitment, and speculation.
Minor:
1- Nobody knows who Nakamoto is/are, so I would not call them a 'scholar'.
Quality of English is mostly fine. I didn't detect any big errors.
Author Response
We added the importance and application of DPoS in Introduction section.
We added literature comparisons in Section 5.3.
Figure 6 is just a flowchart of formulas for updating node's credit value, which is one of the three contributions. Of course, we enlarge the figures for the reader's convenience.
In the conclusion, we presented the limitations of the paper, as well as future improvements to improve the performance of the blockchain. Nodes that successfully generate blocks can receive tokens as rewards to attract more users to participate in.
We used person instead of scholar.
Reviewer 4 Report
The paper presents a contemporary and pertinent investigation of consensus algorithms within the domain of blockchain technology, with a focus on enhancing the Delegated Proof of Stake (DPoS). By introducing a CD-DPoS (Community Discovery-Delegated Proof of Stake), you have taken an ingenious approach to addressing known issues within existing DPoS algorithms. By utilising PageRank and GN algorithms, you provide a multifaceted solution that demonstrates promise for enhancing decentralisation and preventing malicious node activity. Nonetheless, there are a few areas that could be improved:
A literature review that includes more recent studies will provide context and demonstrate a thorough understanding of the current state of the field. It may also assist in identifying any gaps that the CD-DPoS is attempting to fill.
The study would benefit from more explicit details regarding the methodologies employed, such as the node credit value reward and punishment mechanism. A thorough explanation of the procedure and methodology would enhance the scientific rigour of the paper.
Explain the Integration of Algorithms: A more in-depth explanation of how the PageRank and GN algorithms were adapted to the context of DPoS would be beneficial. Explaining how they contributed to the achievement of "one ballot, multiple votes" and measuring the voting enthusiasm of nodes would improve readers' comprehension.
Some sections of the abstract could be reorganised to improve their readability and comprehension. To enhance the English language quality, it is recommended to hire professional editing services or seek peer review.
Follow Template Instructions: Ensure that the paper adheres to the specific formatting and structural guidelines of the journal or conference to which it is being submitted. Adherence to the required template increases the paper's professionalism and likelihood of acceptance.
Author Response
We added literature comparisons in Section 5.3.
We outlined the procedure for credit incentives in Section 3.6.
We explained in depth why the PageRank algorithm is suitable for "one ballot, multiple votes" for CD-DPoS in Section 3.3. In Section 3.4, we also describe why Q is used as a measure of voting enthusiasm in the GN algorithm.
We made appropriate changes to the Abstract section. Actually, this article has used the polish of AJE Professional Agency.
Reviewer 5 Report
A good and original work on improving the DPoS protocol is presented. The reasoning presented in the paper is supported by theoretical calculations and experimental results in practice.
It is recommended to caption Figure 5, on the same page as the figure.
Author Response
We added the title of Figure 5.
Round 2
Reviewer 2 Report
The recommendations were not fully implemented. They should be implemented in full.
The English could be improved.
Author Response
First of all, thanks the editors and reviewers for their hard work. Secondly, we are deeply sorry for the dissatisfaction brought to the experts by the previous revision. In response to the questions proposed by the review experts, we re-examined and made the modifications as much as possible.
We have made some improvements throughout the paper, including reorganized the end of the Introduction section and carefully checked the brackets used in this paper.
We have reorganized the end of page 17 and other similar parts of the paper. In addition, we have focused on alignment issues.
We have carefully checked the expression of sentences in the paper and made some improvements to the best of our ability.
We have added the word "blockchain" as a keyword.
We have added the impact of blockchain on people in the second paragraph of the Introduction section and added some references to the representative literature.
Reviewer 3 Report
The response from the authors is rather cryptic. I provided very specific feedback and recommendations which would require some investment in the paper, but the changes the authors have made are rather small. It is unclear from their 7-sentence response how they tackled each of my previous comments. My previous concerns about the framing of the paper within the governance literature, in the introduction and the literature review, have not been tackled appropriately.
OK
Author Response
First of all, thanks the editors and reviewers for their hard work. Secondly, we are deeply sorry for the dissatisfaction brought to the experts by the previous revision. In response to the questions proposed by the review experts, we re-examined and made the modifications as much as possible.
1- In the third and fourth paragraphs of the introduction, we have added the research of blockchain governance and the importance and application of DPOS respectively, and added references to some literature.
2- We have added a comparative literature analysis work in Section 2.1, the details were shown in Table 1. In addition, we have added references to the above three papers to the Introduction.
3- Actually, Figure 6 is a graph of calculation rules for updating the node credit value, which is one of the three contributions, and we have clearly explained the graph in the third paragraph of Section 3.6. In addition, we have also made changes to the title of Figure 6. In addition, for the convenience of readers, we have enlarged Figure 6.
4- We have presented the limitations of this paper in the Conclusion section and suggested future improvements in terms of voting strategies and communication structure.
5- We have used person instead of scholar.
Round 3
Reviewer 2 Report
This has reached a good level now. Well done.
Reviewer 3 Report
The authors have done a good job answering my initial concerns.